# Novel Poly(Methyl Methacrylate) Containing Nanodiamond to Improve the Mechanical Properties and Fungal Resistance

**DOI:** 10.3390/ma12203438

**Published:** 2019-10-21

**Authors:** Utkarsh Mangal, Ji-Yeong Kim, Ji-Young Seo, Jae-Sung Kwon, Sung-Hwan Choi

**Affiliations:** 1Department of Orthodontics, Institute of Craniofacial Deformity, Yonsei University College of Dentistry, Seoul 03722, Korea; utkmangal@yuhs.ac (U.M.); KATEKIM826@yuhs.ac (J.-Y.K.); JYSEO13@yuhs.ac (J.-Y.S.); 2Department and Research Institute of Dental Biomaterials and Bioengineering, Yonsei University College of Dentistry, Seoul 03722, Korea; 3BK21 PLUS Project, Yonsei University College of Dentistry, Seoul 03722, Korea

**Keywords:** nanodiamond, poly methyl methacrylate resin, nanocomposite, flexural strength, elastic modulus, Vickers hardness, fungal resistance

## Abstract

Herein we evaluate the effect of nanodiamond (ND) incorporation on the mechanical properties of poly(methyl methacrylate) (PMMA) nanocomposite. Three quantities of ND (0.1, 0.3, and 0.5 wt.%) were tested against the control and zirconium oxide nanoparticles (ZrO). Flexural strength and elastic modulus were measured using a three-point bending test, surface hardness was evaluated using the Vickers hardness test, and surface roughness was evaluated using atomic force microscopy (AFM), while fungal adhesion and viability were studied using *Candida albicans*. Samples were also analyzed for biofilm thickness and biomass in a saliva-derived biofilm model. All groups of ND-PMMA nanocomposites had significantly greater mean flexural strengths and statistically improved elastic modulus, compared to the control and ZrO groups (*P* < 0.001). The Vickers hardness values significantly increased compared to the control group (*P* < 0.001) with 0.3% and 0.5% ND. ND addition also gave significant reduction in fungal adhesion and viability (*P* < 0.001) compared to the control group. Finally, salivary biofilm formation was markedly reduced compared to the ZrO group. Hence, the incorporation of 0.1–0.5 wt.% ND with auto- polymerized PMMA resin significantly improved the flexural strength, elastic modulus, and surface hardness, and provided considerable fungal resistance.

## 1. Introduction

Poly(methyl methacrylate) (PMMA) is one of the most common polymers used in the manufacture of a wide range of dental appliances, from interim prosthesis to removable orthodontic retainers, functional appliances, occlusal splints for temporomandibular joint therapy, and different types of surgical splints in craniofacial surgical management. Such versatility of application stems from the ease of manipulation, stability in the oral environment, lack of cytotoxicity, low cost, aesthetic appearance, and color matching of PMMA [1].

PMMA has also acted as a primary model to influence the development of other materials for use in dentistry; however, the inherent mechanical properties of PMMA appliances and products remain less than ideal [2].

The improvement of the mechanical properties of PMMA continues to be one of the actively researched domains in dental biomaterials. Various attempts have been made to modify and improve the mechanical properties by using different metal oxide fillers (aluminum, silver, zirconium, titanium), and fibers (glass, aramid, and carbon graphite fibers) [3,4,5,6,7].

Bond formation and interaction during polymerization between the additive agent and the core polymer must be strong to enhance the mechanical properties against various forces. This is influenced by the ability to achieve high rate and uniformity of dispersion of the modifier particle in the base polymer material [8].

A recent significant development in science and technology has been the wide-scale production and application of nanomaterials, and these are being increasingly adapted in dental materials. The use of various nanoparticles (e.g., quartz, colloidal silica, zirconia, zinc oxide, carbon base derivatives) in dental materials has led to the evolution of materials with notable improvements in chemical, biological, and physical properties [9,10,11].

The use of nanocomposite polymers can offer superior properties to the bulk polymer [12]. This feature has been gaining impetus with the use of ultrafine dispersed diamonds or detonation nanodiamonds (ND), because these nanoparticles exhibit the exceptional properties of bulk diamond, such as high strength, chemical stability, thermal conductivity, and good biocompatibility [13,14,15].

Maximizing the effects of adding nanofillers, recent studies have focused on improving the mechanical characteristics of PMMA using nanoparticles of metal oxides and carbon-based nanofillers [10,11]. Although there is an increasing trend in the application of ND with polymers [16], the application of the same in auto-polymerizing base polymers has been rare [17]. Furthermore, the additional steps required for preliminary surface characterization and predisposition to high agglomeration have been relative hindrances in early regular adaptation of the material [18,19,20,21].

Multi-species biofilms predominantly containing the fungus *Candida albicans* are readily formed on the surfaces of denture bases and orthodontic retainers, resulting in increased susceptibility to stomatitis [22]. Various fillers like platinum, glass fibers, quaternary ammonium compounds, and oxides of silver, titanium, zinc, and zirconia, have been incorporated to address this issue. However, with the increase in microbial resistance, often less or no predictable improvement in the mechanical properties has been observed [23]. Thus, in this study, an in vitro colony culture and simulated salivary biofilm model was designed to analyze the resistance to fungal growth using ND as nanofillers.

Therefore, the focus of the present research was on utilization of ND and incorporation into self- polymerizing orthodontic base polymers.

Here we focused on integrating ND particles at three different concentrations with efficient dispersion into PMMA matrix to enhance the inherent properties of the resin polymer. The samples were evaluated for mechanical properties and biological activity against the conventional orthodontic polymer base as the control. Based on the results of the latest studies [9,10,24,25], zirconium oxide nanoparticles (ZrO) were identified as the positive control. Our null hypothesis was stated as follows: (1) ND incorporation into orthodontic base polymer will not change the mechanical properties, (2) the mechanical properties will be at par or below those of the positive control, and (3) there will be no marked difference in the biological activity between ND-PMMA and the control or positive control groups.

## 2. Materials and Methods

### 2.1. Materials

A commercially available, auto-polymerizing orthodontic acrylic resin system (Ortho-Jet; Lang Dental Manufacturing Co. Inc., Wheeling, IL, USA) was used for all the experiments. ZrO (<100 nm, 1314-23-4; Sigma-Aldrich, Co., St Louis, MO, USA) and ball-milled anhydrous and dispersed NDs (NanoAmando^®^) as fumed diamond powder (3.9–4.3 ± 0.8–1.0 nm) were obtained from the NanoCarbon Research Institute, Ltd., Nagano, Japan.

### 2.2. Silanization of ZrO Particles

To improve the interaction between the resin and the nanoparticles, silanization of ZrO was conducted using silane coupling agent (3-trimethoxysilyl propyl methacrylate, 97% (TMSPM), Sigma-Aldrich, Co., St Louis, MO, USA). The process of coating involved dissolving 0.3 g of TMSPM in 100 mL of acetone, followed by adding 30 g ZrO to the TMSPM/acetone mix and stirring for 60 min with a magnetic stirrer (Cimarec Digital Stirring Hotplates, SP131320-33Q; Thermo Fisher Scientific, Waltham, MA, USA), as previously reported [26]. 

When the sample had dried, it was heated at 120 °C for 2 h and then naturally cooled to obtain the surface-treated ZrO sample, which was further cooled to obtain silanized ZrO [27].

### 2.3. Preparation of the Nanocomposite Specimens

The ND-PMMA composite was fabricated by mixing the self-polymerizing resin powder with a solution of monomer and ND particles. ND was pre-weighted to 0.1, 0.3, and 0.5 wt% with respect to the powder. The particles were then magnetically stirred (IKA Werke RT 15 Power 15 Position Stirrer, IKA^®^ Korea. Ltd, Seoul, Korea) at 1000 rpm for 15 min, followed by ultrasonication in a cold water bath (SH-2100; Saehan Ultrasonic, Seoul, Korea) for 15 min to avoid excessive heating of the liquid. The samples were then allowed to cool while being magnetically stirred continuously for a further 10 min (Figure 1).

The ZrO composite with resin matrix was prepared by mixing the 5 wt.% of the silanized ZrO particles with resin powder. The oxide nanopowder was thoroughly mixed with the acrylic polymer powder using a porcelain mortar and pestle, and then, the monomer was added to the mixture [24].

In previous studies [10,24,28], the reinforcement with 5 wt.% ZrO provided marked improvement in the mechanical properties of the base polymer; therefore, in this study, 5 wt.% ZrO-nanocomposite was set as the positive control. All the specimens were prepared at an optimum ratio of resin powder to liquid (2.5:1.5) as per the literature [17]. After low-temperature polymerization (60 °C, 4.0 bar, 15 min, Air Press Unit, Sejong Dental, Daejeon, Korea), specimens of different shapes were obtained from each experiment and polished with silicon carbide papers up to 2000 grit. The cured specimens were stored in distilled water at 37 °C for 48 h prior to each experiment.

### 2.4. Characterization of the Nanoparticles and the Samples

The characterization of the ZrO was done using field-emission scanning electron microscopy (FE-SEM; JEOL-7800F, Tokyo, Japan) for evaluation of effective silanization and particle size. The ND particles were analyzed for dispersion and size in the monomer liquid by transmission electron microscopy (TEM) using a JEM-200F microscope (JEOL, Tokyo, Japan) operated at 200 kV.

### 2.5. Fracture Surface Analysis

To characterize the nano-polymerized specimens, 5 wt.% ZrO and 0.1%, 0.3%, and 0.5% ND-incorporated PMMA samples, and the control sample, with dimensions of 3.3 mm × 10 mm × 25 mm, were fractured by a computer-controlled universal testing machine. The fractured surfaces of the ND-PMMA samples were coated with Pt (5 nm), and the ZrO sample was gold coated, using an ion coater (ACE600; Leica). The samples were photographed via FE-SEM (Carl Zeiss, Oberkochen, Germany) at 2 kV.

### 2.6. Surface Characterization and Hydrophilicity

Surface irregularities form a conducive niche for the proliferation of microorganisms and prevent efficient cleansing solely with mechanical sheer forces. The assessment of surface characteristics can be made by evaluating the roughness and the response to the sessile water drop contact angle. The surface roughness of each sample (n = 2) was analyzed by atomic force microscopy in a scan area of 5 μm × 5 μm (AFM, Nanowizard I, JPK Instruments AG, Berlin, Germany). Arithmetic roughness (Ra) was measured, and the values from three measurements of each specimen were averaged [29]. To evaluate the hydrophilicity of each sample (n = 5), we assessed the contact angle of a 5-μL droplet of dH_2_O placed at a rate of 2.0 μL·s^−1^ at the center of each sample surface at room temperature. The contact angles were measured by a droplet analysis device (SmartDrop, Femtofab, Korea) using the sessile drop method [30,31].

### 2.7. Mechanical Testing

The PMMA-based appliance in dentistry experiences various tensile, compressive shear, and abrasive forces during its function. Mechanical testing enables the evaluation of the ability of the material to withstand such forces and describes its elastic nature. In the present study, the mechanical properties were analyzed following the specifications of ISO 20795-2 [32]. Five groups of samples were fabricated each with n = 10 and dimensions of 3.3 mm × 10 mm × 25 mm. A computer-controlled universal testing machine (Model 3366; Instron^®^, Norwood, MA, USA) was used to fracture the specimens in three-point flexure. The flexural strength and elastic modulus were measured at a span length of 50 mm and a crosshead speed of 5 mm/min. The flexural strength and elastic modulus were calculated as per the standard equations [32]. 

A Vickers hardness machine (DMH-2, Mastsuzawa Siki Co. Ltd., Akita, Japan) was used to determine the hardness (HV) with 50 g force (0.49 N) for 30 s; the average value was calculated from readings taken at three different locations on each specimen [33].

### 2.8. Microbial Attachment, Colony-Forming Unit, and Viability

The microorganisms were obtained from the Korean Collection for Oral Microbiology (KCOM, Gwangju, Korea) and tested against the disc shaped specimens with a diameter of 10 mm and thickness of 2 mm, formed using a mold. Fungal interaction was analyzed using *Candida albicans* (KCOM 1301), cultured in a YM broth (Becton Dickinson and Co., Sparks, MD, USA). To evaluate the colony-forming units (CFUs), disc samples with yeast cell attachments were washed in 1 × Phosphate-buffered saline (PBS) to remove nonadherent cells. Cells were resuspended in 1 mL of distilled water by sonicating (SH-2100; Saehan Ultrasonic, Seoul, Korea) for 5 min, followed by pipetting up and down. Of this microbial suspension, 100 µL was spread onto the respective agar plates and incubated at 37 °C for 24 h, after which the total number of colonies were then counted using the ImageJ/Fiji software (version 1.52av for Windows, Java 1.8.0_112 64-bit) with a customized macro [34]. 

The viability of the adherent fungal cells was determined by staining using a live/dead viability kit (Molecular Probes, Eugene, OR, USA), according to the manufacturer’s protocol. Equal volumes of Syto 9 dye and propidium iodide, which stain live (green) and dead (red) organisms, respectively, from the kit, were mixed thoroughly. A 3-µL aliquot of this mixture in PBS was used to stain the samples. The stained samples were incubated at room temperature, in the dark for 15 min, and were then observed with a confocal laser microscope (CLSM, LSM700; Carl Zeiss, Thornwood, NY, USA).

### 2.9. Saliva-Derived Biofilm Model and Biomass Measurement

According to McBain, in order to grow the plaque microcosm with its complexity and heterogeneity akin to in vivo conditions, human saliva offers the most ideal and conducive environment [35]. Adapting the protocol from literature [36], human saliva was collected from healthy adult donors who had no active caries or periodontal disease and had not taken antibiotics within the past 3 months. Participants did not brush their teeth for 24 h and abstained from oral intake for at least 2 h prior to donating saliva. The saliva was obtained in accordance with the ethical principles of the 64th World Medical Association Declaration of Helsinki and following procedures approved by the institutional review board of the Yonsei University Dental Hospital (Seoul, Korea). Written informed consent was obtained from all participants before they donated saliva. Saliva was collected from six individuals and mixed in equal proportions. The mixed saliva was then diluted in sterile glycerol to a concentration of 30% and stored at −80 °C to be used as a biofilm model [37].

The biofilm model was cultured in McBain medium supplemented with mucin (type II, porcine, gastric) (2.5 g/L), bacteriological peptone (2.0 g/L), tryptone (2.0 g/L), yeast extract (1.0 g/L), NaCl (0.35 g/L), KCl (0.2 g/L), CaCl2 (0.2 g/L), cysteine hydrochloride (0.1 g/L), haemin (0.001 g/L), and vitamin K1 (0.0002 g/L), at 37 °C for 24 h [35,38]. From the cultured medium, 1.5 mL of the microbial solution was placed on the specimen. Following 8 h, 16 h, and 24 h of incubation, an additional 1.5 mL of microbial solution was placed on the specimen, after each period. Biofilms could grow for a total of 48 h. Specimens were stained using the live/dead cell viability kit (Molecular Probes, Eugene, OR, USA) via a method similar to that used for yeast cell viability. 

Images of the axially stacked biofilm were captured using CLSM (LSM700, Carl Zeiss, Thornwood, NY, USA) and the thickness of the biofilm was calculated using the software ZEN (Carl Zeiss, Thornwood, NY, USA). Additionally, the COMSTAT plug-in for the ImageJ (NIH, Bethesda, MA, USA) software was used to determine the average biomass [39,40].

### 2.10. Statistical Analysis

All statistical analyses were performed using IBM SPSS software, version 23.0 (IBM Korea Inc., Seoul, Korea) for Windows, with data from at least three independent experiments. The results obtained from the control and experimental groups were analyzed by one-way analysis of variance (ANOVA) followed by Tukey’s test. *P* < 0.05 is considered statistically significant.

## 3. Results

### 3.1. Characterization of Nanoparticles

In FE-SEM characterization, ZrO particles were visualized as silanized particles with sizes in the range of 25–50 nm and marked tendencies towards aggregation. (Figure 2A,B). The TEM scanning for ND particles showed particles with sizes in the range of 4–6 nm and tendencies towards agglomeration (Figure 2C,D).

### 3.2. Surface Characteristics

#### 3.2.1. Scanning Electron Microscope (SEM)

The fractured surface of pure PMMA specimens displayed a smooth surface. The fractured surface of the ZrO composite showed signs of cracks, high particle clustering with multiple voids, and revealed a ductile type failure behavior exhibiting an irregular and rough surface. The ND groups presented more clear fracture features and nanoparticle aggregation in several places and small voids with the increase in ND content (Figure 3).

#### 3.2.2. Surface Roughness

With the AFM analysis (Figure 4A), the ZrO nanocomposites (Ra: 207.533 ± 97.48 nm) recorded the highest average roughness, and the control group (Ra: 146.792 ± 22.06 nm) showed the lowest value. Among the ND groups, the groups containing 0.1% ND, 0.3% ND, and 0.5% ND had Ra values of 162.502 ± 4.24 nm, 176.258 ± 4.51 nm, and 163.261 ± 18.68 nm, respectively. However, test groups and control groups showed no statistically significant differences (Figure 4B).

#### 3.2.3. Hydrophilicity

The wettability of the samples was evaluated by measuring the water contact angle on their surfaces (*P* < 0.001, Figure 5A). The average value of the contact angles was observed to be the highest for the control group (65.819° ± 2.98°) and lowest for the ZrO nano-composite group (56.07° ± 6.35°), while the contact angle for the ND groups showed a downward trend. The average angle measures showed statistically significant differences in ZrO nanocomposites in the control group and 0.1% ND group; however, there was no significant difference between the control group and 0.1% ND group.

### 3.3. Mechanical Properties of Nanocomposites

#### 3.3.1. Flexural Strength and Modulus

The ND test groups showed a significant increase in the flexural strength in comparison to the control and ZrO group. (*P* < 0.001, Table 1, Figure 5B). The modulus of elasticity also showed significant differences between control and ND groups (*P* < 0.01, Figure 5C). In the ND test groups, flexural strength of the 0.1% ND group was higher than that of the 0.3% ND and 0.5% ND groups; whereas the elastic moduli of the ND groups showed the opposite trend on increasing the ND content. 

However, the differences in both the parameters were not statistically significant among the ND groups.

#### 3.3.2. Hardness Test

The graphical representation of the Vickers hardness (HV) assessment shows that the hardness value improves with the increasing weight percentage of NDs (*P* < 0.001, Figure 5D). The hardness values for control (12.89 ± 1.81 kg/mm^2^) were significantly different from ZrO (14.327 ± 1.38 kg/mm^2^), 0.3% ND (14.487 ± 1.51 kg/mm^2^), and 0.5% ND (15.653 ± 1.223 kg/mm^2^); although the values for the control were comparable to 0.1% ND (13.860 ± 1.27 kg/mm^2^), there was still a significant difference.

### 3.4. Microbial Attachment, Colony Forming Units, and Viability

The confocal imaging showed noticeable differences between the groups. The viable yeast cells attached to the samples were stained with a green fluorescent stain; these cells were found to be greater in number for the ZrO nano-composite group than for the ND (Figure 6A). This correlated to the quantitative analysis conducted for the count of CFUs for *Candida albicans* in ND groups, which showed significantly lower CFUs for ND test groups than that of the ZrO group (*P* < 0.001, Figure 6B). 

### 3.5. Biofilm Thickness and Biomass

The results showed distinctive differences between the control and ND groups; this is consistent with those observed with a single microbial analysis (Figure 7A). The biofilm thickness was observed to be highest in the ZrO composite and significantly less in ND-composite groups; these results were subsequently confirmed with the software, which also revealed statistically significant differences (*P* < 0.05, Figure 7B). Finally, the biomass of the biofilm trends towards reduction with an increase in concentration of ND; however, the reduction was not statistically significant (Figure 7C).

## 4. Discussion

The widespread application of PMMA in dentistry has driven biomaterial research towards overcoming underlying inadequacies such as low mechanical strength and hardness. The advent of nano sized zirconia dioxide particles has further advanced this domain with multiple studies that were aimed to achieve enhanced materials with the conjugation of PMMA and ZrO [11,27,41,42,43]. This has particularly been made possible due to good biocompatibility of zirconia in addition to its inherent property of high hardness [24,44]. Over the past decade, with advancement in manufacturing, the availability of nano derivatives of carbon such as nanodiamond has increased. NDs mimic the inherent superior physical properties of bulk diamond and have recently been demonstrated to be safe and biocompatible in multiple studies [45,46,47,48]. Therefore, the present study was designed to assess the improvements in the properties of poly(methyl methacrylate) on the addition of three different weight percentages of ND as nanofillers, with ZrO filler as the positive control.

In augmenting the mechanical properties of the polymer with a filler, it is essential to achieve optimal to high interaction between the individual constituent units. However, the structure of the nanofiller is the deciding factor when utilizing zirconium oxide, in that nanopowder offers marked improvement in properties upon modification by silanzation [27]. Therefore, in the current study, the ZrO was silanized using TMSPM and the results were affirmed using FESEM-EDX.

Numerous studies have advocated additional surface characterization for the ND when used as a carrier for drug or active binding [14,15,16]. However, as the ND undergoes multiple stages in the manufacturing process, it experiences changes in the surface chemistry. This process has often been referred to as “self-functionalization” [49]. The focus of the present study was to use the ND as a nanofiller only; therefore, the dried ND powder was used in the same state as it was supplied, according to the manufacturer’s instructions. Furthermore, a homogenous distribution of NDs in the polymer matrix is imperative to improve the performance of the nanocomposite [50]. Therefore, the ND powder was mixed at a high speed and sonicated with the monomer liquid prior to the polymerization reaction.

Surface roughness is one of the key determinant factors in microbial colonization. An average roughness (Ra) value of more than 0.2 μm leads to an increase in microbial attachment [51]. In the present study, the analysis of the nanocomposites with AFM showed average values <200 nm.

The wettability of the surface is characteristic of the surface energy. The surface energy of NDs is inherently low, and in the present study, the 0.1% ND nanocomposites showed higher contact angles compared to ZrO. However, the contact angle reduced with an increase in the concentration of the NDs, which might be due to the tendency of agglomeration of ND particles and reduction in homogeneity within the matrix.

High flexural strength forms an essential requirement of the base polymer material, even from the point of standardization [32]. With the use of ZrO fillers, the majority of the studies in the past have focused on heat cured acrylic resin [11,27,43,52]; a few studies have assessed auto-polymerized resin [10,24] with no distinct focus on orthodontic base polymers. The results for flexural strength and elastic modulus in the present study were found to be similar to that of previous studies with chemically cured resins, conducted by Alhavaz et al. [24]; these showed a mild improvement in mechanical strength when compared to the control with ZrO. However, the results were noticeably different from the earlier studies that used heat cured resin; this highlighted the variability in the outcome with application of the same nanofillers employing two different polymerization methodologies. Furthermore, the reduction in the flexural strength can be attributed to the high agglomeration tendency depicted by the ZrO particles as observed in FE-SEM imaging. This is also believed to have resulted in the generation of multiple voids that were visualized in the SEM analysis, vitiating the resistance to loading forces. 

High surface hardness for resin polymers is another desirable characteristic to resist surface abrasion. Studies in the past were conducted to enhance this property with the addition of metal oxides in the base polymers and these results are promising [52,53,54]. In the present study, the addition of ZrO exhibited significant improvement on the result of surface hardness, which is in agreement with the previous studies [24,33]. This improvement with metal oxide addition is attributed to the possibility of strong interatomic ionic interactions in the matrix [24,53].

The rich surface chemistry and superior mechanical properties of nanoscale diamond particles make them an excellent filler material for composites [55]. However, the prerequisite to achieve homogenous dispersion in the matrix presents a caveat, because without such dispersion, the thermal and mechanical properties would degrade [15]. While earlier studies [56,57] have reported on improvements in the mechanical properties of epoxy resin with ND as fillers, there is very limited research on the application of ND together with PMMA. Protopapa et al. [17] reported enhancements in the mechanical properties of auto-polymerized PMMA with the addition of ND. In this study, the elastic modulus values are concordant but the flexural strength variation values are contrasting. Further, an increase in ND quantity from 0.1 to 0.5 wt.% did not decrease the mechanical performance markedly, and all proportions showed significantly higher improvement in flexural strength.

Al-Harbi et al. [21] evaluated the use of ND powder filler in heat polymerized resin. However, the ND quantities used in the test groups were 0.5, 1, and 1.5 wt.%. While they reported a significant increase in strength with 0.5% filler, deterioration was seen with use of as high as 1.5 wt.% of ND. Similarly, in this study we observed a significant increase in strength with the use of 0.5 wt.% ND.

This pattern of changes can potentially be attributed to two characteristics: the agglomeration of ND particles in the polymer matrix and the homogeneity of dispersion of the particles. Homogenous distribution enables uniform interparticle interaction; however, an increase in the amount of ND can lead to higher localized agglomeration [16,58].

The interfacial interaction between filler and matrix is also an important factor determining the mechanical properties of polymer nanocomposites. Significant increase in surface hardness with the addition of ND has been previously reported with the use of ND with different surface chemistry [59]. The results of this study also agree with these previous results. 

The use of PMMA base polymers is intended for an oral environment containing several microorganisms. The initial adherence of *Candida albicans* is influenced by the surface characteristics resulting from the chemistry of the polymer. This initial adherence to the resin creates an environment for further bonding and colonization. In this study, the ND nanocomposite samples showed higher resistance to fungal attachment, tested with *Candida albicans* in comparison to the control. However, in contrast to a previous study [41], the ZrO composite samples in this study were negatively affected with an increase in yeast cell attachment and proliferation, observed via count and imaging. The interaction of ND with bacteria and fungi has been reported to be multifactorial, depending on the size, shape, distribution, and concentration of the ND [60,61]. In the present study, with the addition of ND, significant improvement in the fungal resistance was observed. The predominantly fungistatic effect observed in previous studies has been attributed to the positive zeta potential of ND. The surface charge characteristic of ND impacts growth without cellular damage, as per Chwalibog et al. [62]. Further, with the addition of ND, the hydrophilicity of the nanocomposite increases. This increase has been found to be related to the formation of a hydration layer that can prevent the adhesion of microorganisms such as *Candida albicans* [63]. 

Finally, we evaluated the thickness and biomass of the multi-species biofilm formed in the complex salivary ecosystem and analyzed it using CLSM images. The results revealed that both the thickness and biomass of the biofilm formed on the ND nanocomposites were considerably lower than those in the control. In addition, the ZrO group showed a marked increase in the biomass and biofilm thickness, maximum amongst all the test groups. This can be attributed to the high surface roughness of the ZrO group (Ra: 207.533 ± 97.48 nm) beyond the clinically recommended value of 0.2 μm, as per Bollen et al. [64]. 

Though there was a reducing trend for both cell culture and biofilm analysis with increasing weight percentage of ND, further studies with different microorganisms are required to ascertain the interaction between ND-PMMA polymer and oral microbes. Nonetheless, the results of this study positively indicate a potential for future application.

Further, with the addition of nanofillers, a change in color and transparency were observed. Whereas in the case of ZrO fillers, there was a loss of transparency, there was only a mild but incremental loss with increase in the ND content from 0.1 wt.% to 0.5 wt.% (image on the bottom left in Figure 1).

The results of this study address the major physical and limited biological properties that manifest solely in an in vitro environment. Therefore, further research to assesses additional biological parameters, for clinical application, is essential. Thus, the results should be interpreted carefully and be further developed for direct use in the future.

## 5. Conclusions

Within the limits of the study, the reinforcement of poly(methyl methacrylate) (PMMA) polymer with ND resulted in significant improvement in the mechanical properties, with the use of as little as 0.1 wt.% ND, resulting in more than 20% increment in flexural strength over unmodified PMMA. In addition, ND incorporation showed marked resistance to *Candida albicans* and considerable reduction in salivary biofilm formation.

## Figures and Tables

**Figure 1 materials-12-03438-f001:**
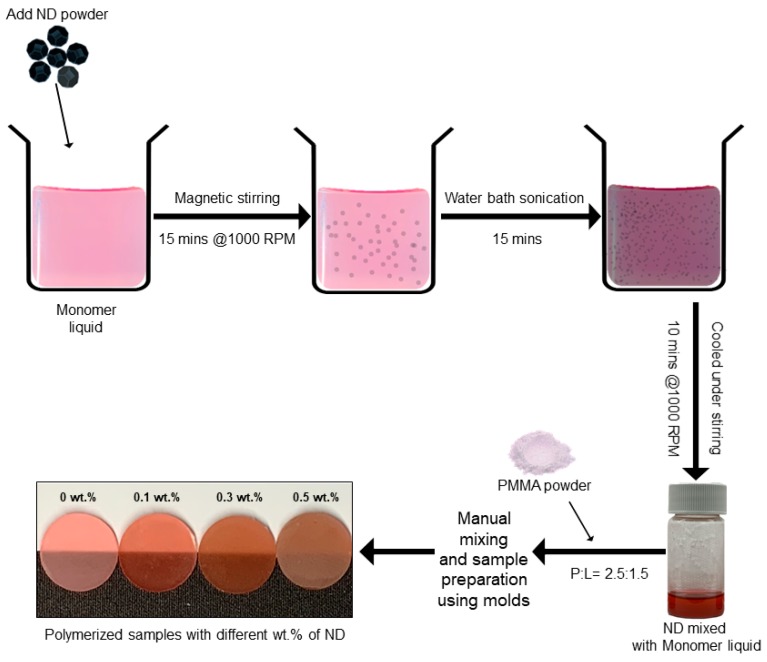
Schematic of the preparation of nanodiamond(ND)-PMMA nanocomposite.

**Figure 2 materials-12-03438-f002:**
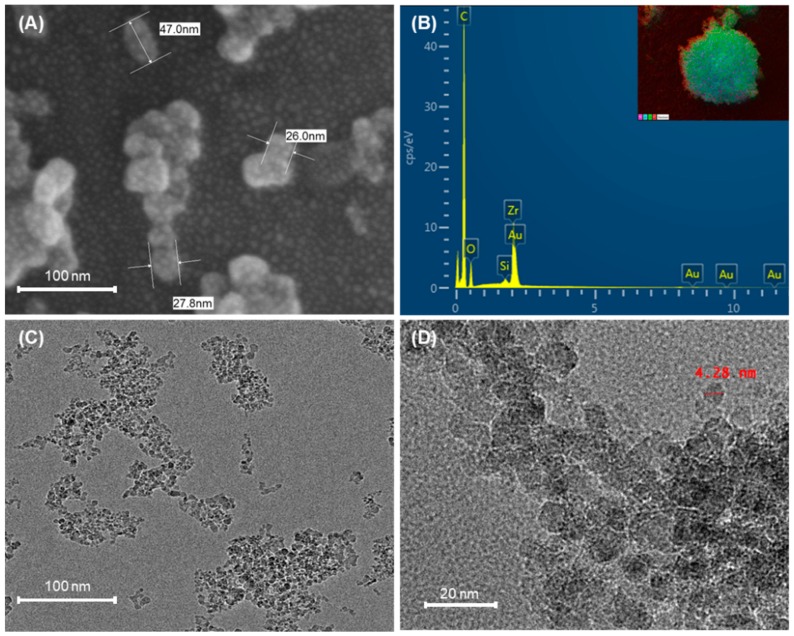
Characterization of the nanoparticles. (**A**,**B**) FE-SEM images of particle size and shape distributions and the corresponding EDX spectrum of silanized ZrO; (**C**,**D**) TEM images of ND particles displaying the nanometer dimensions in the range of 4–6 nm.

**Figure 3 materials-12-03438-f003:**
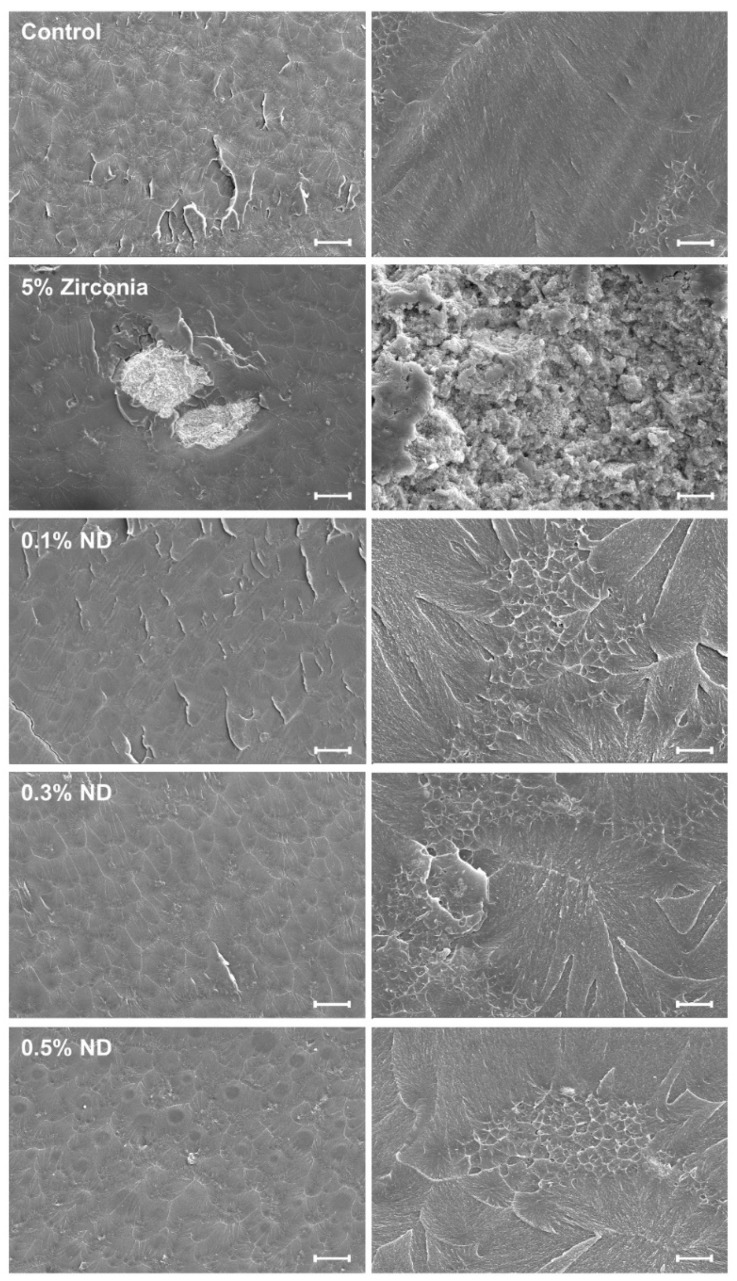
Representative SEM images of fractured surfaces of specimens. Images in each row comprise the same group. The scale bar is 200 μm (left) and 2 μm (right), respectively.

**Figure 4 materials-12-03438-f004:**
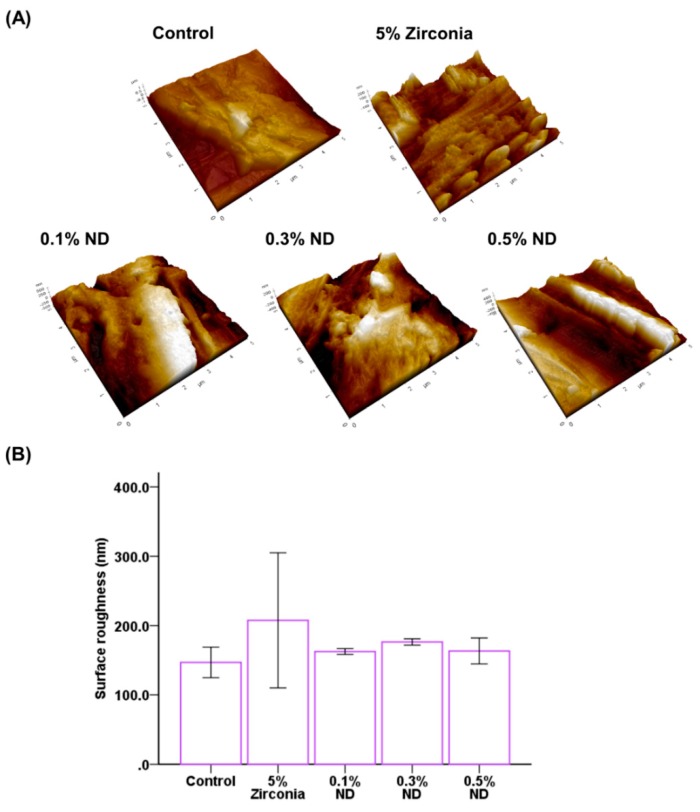
Average surface roughness of different groups of samples using AFM (scan area: 5 μm × 5 μm). (**A**)Three-dimensional surface topographic images; (**B**) Comparison of the average surface roughness of different groups.

**Figure 5 materials-12-03438-f005:**
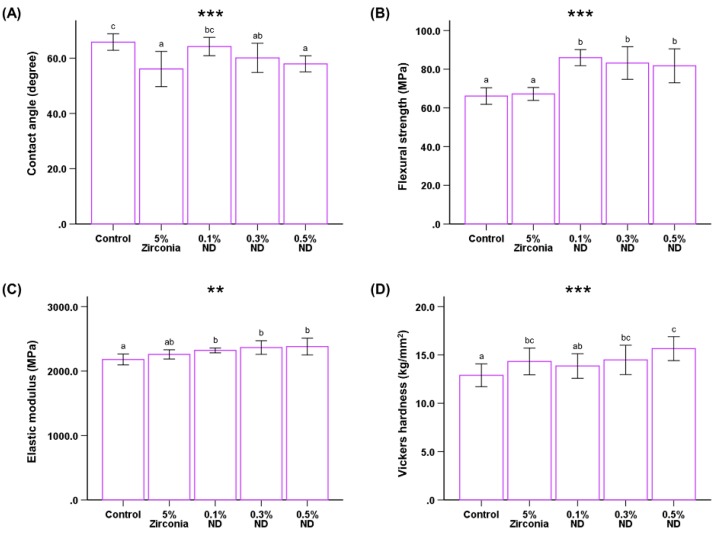
Comparison of the water contact angle (**A**), flexural strength (**B**), elastic modulus (**C**), and Vickers hardness (**D**) among different groups of samples. Different letters above the bars indicate significant differences. ** *P* < 0.01, *** *P* < 0.001 for comparison between the groups.

**Figure 6 materials-12-03438-f006:**
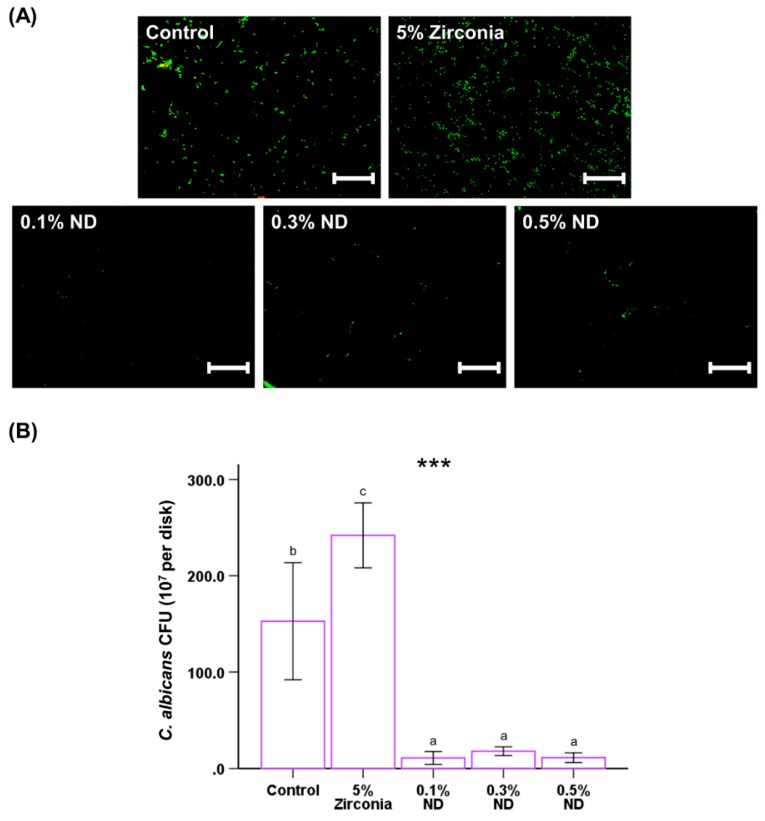
Representative live/dead staining images of fungi attached on the surfaces of poly(methyl methacrylate) (PMMA) with control, different wt.% ND and ZrO as positive control, for *Candida albicans* (**A**). The scale bar is 100 µm. Colony-forming unit (CFU) counts derived from fungi attached on the surfaces of PMMA with different concentrations of ND and 5% ZrO samples (**B**). The different letters above the bars indicate significant differences. *** *P* < 0.001 for comparison between the groups.

**Figure 7 materials-12-03438-f007:**
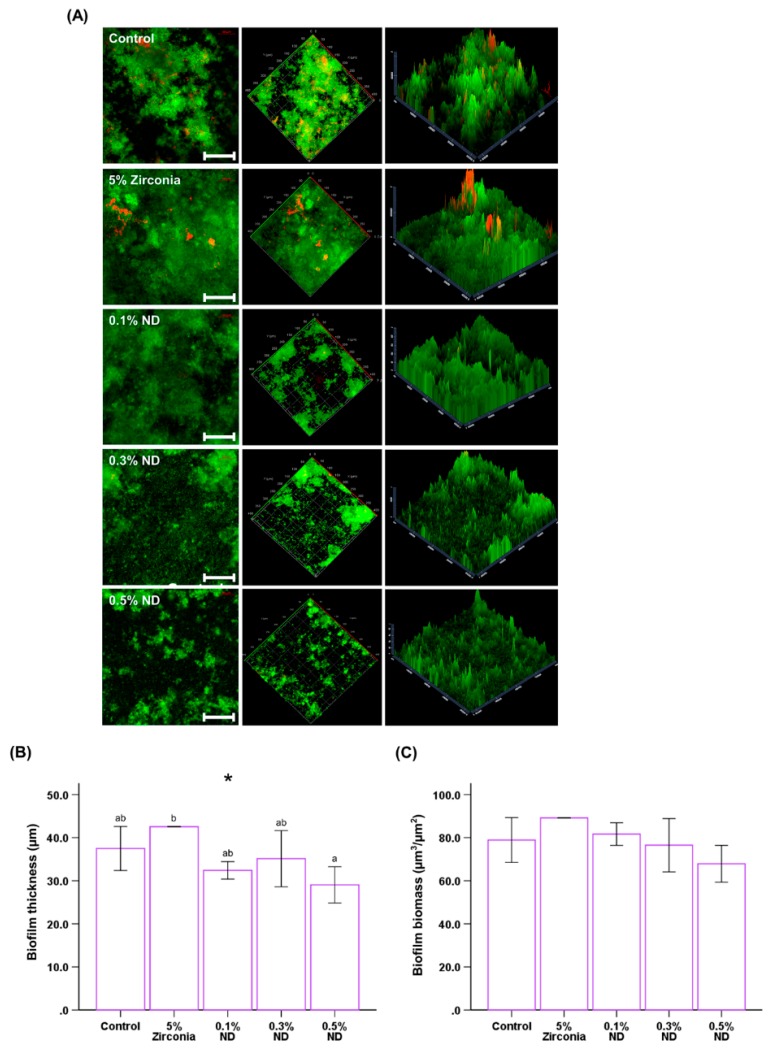
Representative live/dead staining images of biofilms (**A**) attached on the surfaces of PMMA with control, different wt.% of ND and 5% ZrO. The scale bar is 100 µm. Quantitative analysis of the thickness (**B**) and biomass (**C**) of the biofilms. Different letters above the bars indicate significant differences. * *P* < 0.05 for comparison between the groups.

**Table 1 materials-12-03438-t001:** Mean (MPa) and standard deviation (SD) values of flexural strength and elastic modulus for different groups.

Groups	Flexural Strength (MPa) ***	Elastic Modulus (MPa) **
Control	66.12 ± 4.27 ^a^	2179.40 ± 84.15 ^a^
5% Zirconia	67.21 ± 3.32 ^a^	2258.10 ± 71.83 ^ab^
0.1% ND	85.96 ± 4.15 ^b^	2320.25 ± 36.99 ^b^
0.3% ND	83.19 ± 8.43 ^b^	2365.06 ± 105.87 ^b^
0.5% ND	81.75 ± 8.76 ^b^	2379.50 ± 130.19 ^b^

Within a column, cells having similar (lower case) letters are not significantly different. ** *P* < 0.01, *** *P* < 0.001 for comparison between the groups.

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
