# Peer review of "Novel Poly(Methyl Methacrylate) Containing Nanodiamond to Improve the Mechanical Properties and Fungal Resistance"

_materials, 2019, doi:10.3390/ma12203438_

Round 1

Reviewer 1 Report

The study found significant improvement of mechanical and fungal resistances due to addition of ND.

The study reported most of the results nicely except the reason why ND addition improves the PMMA fungal resistance. A further discussion is required.

The paper is too long and can be reduced by removing equations.

Author Response

Reviewer #1

General Comments

The study found significant improvement of mechanical and fungal resistances due to addition of ND.

Response to General Comments

Thank you for your valuable comments and appreciation for the research work.

Major Concerns

The study reported most of the results nicely except the reason why ND addition improves the PMMA fungal resistance. A further discussion is required.

Response to Major Concerns

Thank you for your valuable input. Following the suggestion, the discussion (section 4) has been appended with the rationale behind the possible fungal resistance displayed by the ND-PMMA complex and additional reference is included in support of the same.

Minor Concerns 1

The paper is too long and can be reduced by removing equations.

Response 1

As per the thoughtful suggestion, the section has been formatted after removing the equations.

Reviewer 2 Report

In this paper, the nanodiamond (ND) was used as modifer to reinforce an autopolymerizing orthodontic acrylic resin (PMMA), so that properties of PMMA can be improved for dental applications. The results shows, ND helps by improving some mechanical properties e.g. flexural strength and Vickers hardness. Meanwhile, the fungi resistance of ND-filled compounds was increased significantly.

Generally, the experiments were logically designed and aimed well at the essential practical properties for the dental application. The discussion part was carried outcarefully and convincedly. Eventhough, there are still many places to be better explained or improved:

The English should be checked carefully. E.g. sentences beginning with “the need” sound strange and should be reformed. In the introduction, there should be more backgrounds about the influence of fillers to fungi resistance, as almost half of your results is about the fungi resistance. And the planned research about the fungi resistance should also be highlighted here, because this part makes your paper different from the previous researches, which only focus on the mechanical properties. While introducing the testing methods in section 2, the mean of the testing to practical applications should be mentioned. Otherwise, audience cannot catch the connection between the testing and the purpose of this paper while reading this part. Section 2.3: A graphical sketch would be helpful to quickly understand the preparation of specimens Section 2.7: The definition "l is the distance" is not clearly enough. It should be the length of deformation. The range of strain for evaluating of elastic modulus must be clearly defined for polymer. Section 3.1: The texts in Figure 1 (A) are too small and the scale is missing.   Section 3.3.1: There is some difference of the elastic modulus between control and ND-groups from statistical view. But the percentage difference is actually small (maybe 3-5 %). Thus, you cannot say in the abstract, that the elastic modulus is “significantly improved“. Section 4: Why should the wettability be tested? What’s the connection between the wettability and the dental application? Section 4: You wrote “with addition of the nanofillers, a visible change in color and transparency was observed”. You need to show some evidences for that, such like photos of specimens. You cannot just say it in the discussion part.

Author Response

Reviewer #2

General Comments

In this paper, the nanodiamond (ND) was used as modifer to reinforce an autopolymerizing orthodontic acrylic resin (PMMA), so that properties of PMMA can be improved for dental applications. The results shows, ND helps by improving some mechanical properties e.g. flexural strength and Vickers hardness. Meanwhile, the fungi resistance of ND-filled compounds was increased significantly.

Generally, the experiments were logically designed and aimed well at the essential practical properties for the dental application. The discussion part was carried out carefully and convincedly.

Response to General Comments

The authors would like to thank the reviewer for the interest and in-depth review with valuable and helpful comments for improvement of the manuscript.

Major Concern-1

In the introduction, there should be more backgrounds about the influence of fillers to fungi resistance, as almost half of your results is about the fungi resistance. And the planned research about the fungi resistance should also be highlighted here, because this part makes your paper different from the previous researches, which only focus on the mechanical properties

Response to Major Concern-1

As per the suggestions, the introduction section has been modified to include background on use of the fillers and the microbial interaction. The lacunae in the findings is identified and brief outline of the present study design is also mentioned.

Major Concern2

While introducing the testing methods in section 2, the mean of the testing to practical applications should be mentioned. Otherwise, audience cannot catch the connection between the testing and the purpose of this paper while reading this part.

Response to Major Concern2

The section 2 has been amended as per the suggestions, to highlight the practical applications of the test.

Minor Concerns 1

The English should be checked carefully. E.g. sentences beginning with “the need” sound strange and should be reformed.

Response 1

The language was verified, and modification have been made including the sentence structure as mentioned in the E.g. above.

Minor Concerns 2

Section 2.3: A graphical sketch would be helpful to quickly understand the preparation of specimens

Response2

The steps in preparation of ND-PMMA have been schematically depicted in the figure 1 including the specimen photograph.

Minor Concerns3

Section 2.7: The definition "l is the distance" is not clearly enough. It should be the length of deformation. The range of strain for evaluating of elastic modulus must be clearly defined for polymer.

Response3

Following the suggestions of the Reviewer #1, the section was formatted with elimination of the equations. Therefore, the information is now cited to the reference and elaborated.

Minor Concerns 4

The texts in Figure 1 (A) are too small and the scale is missing.

Response 4

The figure 1 is reordered as figure 2. It has been revised with uniform scale pattern for all the images and formatted to improve the visibility of the text.

Minor Concerns 5

Section 3.3.1: There is some difference of the elastic modulus between control and ND-groups from statistical view. But the percentage difference is actually small (maybe 3-5 %). Thus, you cannot say in the abstract, that the elastic modulus is “significantly improved “

Response 5

The abstract has been modified to correctly depict the improvement in elastic modulus as “statistically improved”

Minor Concerns 6

Section 4: Why should the wettability be tested? What’s the connection between the wettability and the dental application?

Response 6

The dental application of PMMA is in the form of intra-oral appliances. Therefore, the appliances endure prolonged exposure to moist oral environment. The dominant microbial species such as C.albicans, S.mutans etc are hydrophobic in nature or bind preferentially to surfaces with hydrophobic nature. Therefore, the wettability test was done to verify that the addition of the filler does not compromise, if not improve, the properties of the PMMA as nanocomposite is formed. Subsequently, it was observed with sessile contact angle analysis that with increase in ND wt% the hydrophilicity improved in a statistically significant manner, compared to the control group.

Minor Concerns 7

Section 4: You wrote “with addition of the nanofillers, a visible change in color and transparency was observed”. You need to show some evidences for that, such like photos of specimens. You cannot just say it in the discussion part.

Response 7

The statement in concern was independent observation from the tests conducted and has no supportive statistical data. Therefore, as per suggestion the schematic figure included also displays the polymerized samples, supporting the point in concern.